# Effect of Welding Current on Corrosion Resistance of Heat-Affected Zones of HDR Duplex Stainless Steel

**DOI:** 10.3390/ma17091986

**Published:** 2024-04-25

**Authors:** Xin Liu, Yulong Hu, Nian Liu

**Affiliations:** 1Department of Basic, Naval University of Engineering, Wuhan 430033, China; liuxin2008dragon@126.com; 2Navy 91844 Troops, Guangzhou 510033, China

**Keywords:** welding, heat-affected zone, HDR duplex stainless steel, structure, corrosion behavior

## Abstract

This paper examines the corrosion behavior of the welding heat-affected zone (HAZ) of HDR (high chromium, duplex, corrosion-resistant) duplex stainless steel, which currently faces corrosion-related challenges in marine seawater systems. The corrosion behavior of the HAZ was studied using microstructure analysis, polarization curve experiments, and double-loop potentiodynamic reactivation experiments. The results show that (1) the covering welding current can promote the formation of austenite in the HAZ, and that the covering welding current has no strict correspondence with the formation of austenite; (2) increasing the welding gap properly can facilitate the formation of austenite; (3) increasing the covering welding current enhances the material’s pitting resistance, and a covering welding current of 70 A, coupled with a covering welding current of 100 A, represents a reasonable choice in terms of achieving a stronger pitting resistance; (4) in terms of intergranular corrosion resistance, increasing the covering welding current is not conducive to the intergranular corrosion resistance of the material, as the covering current will promote the precipitation of the secondary phase at the grain boundary, thus reducing its intergranular corrosion resistance; and (5) reducing the welding current appropriately contributes to improving the stability of the grain boundary.

## 1. Introduction

Water cooling systems are an important and indispensable part of the production process in many industrial fields, such as ships, electric power, oil refining, the chemical industry, and so on. At present, seawater systems commonly used in ships incorporate materials such as copper alloys. However, due to the inadequate corrosion resistance of copper alloys against seawater flow, corrosion frequently manifests within seawater systems [1,2]. Duplex stainless steel shows excellent performance in handling seawater flow, and its utilization in ship applications is gradually expanding. The corrosion resistance of duplex stainless steel is closely related to its organizational state [3,4,5]. Improper processing, welding, and other treatments can lead to problems such as phase disproportion, the precipitation of harmful secondary phases, and more, thus potentially reducing corrosion resistance and resulting in corrosion-related challenges [6,7,8].

Hence, the corrosion behavior of the welding heat-affected zone (HAZ) of HDR (high chromium, duplex, corrosion-resistant) duplex stainless steel was examined via analyzing the composition and structure of the HAZ, and through conducting relevant corrosion experiments.

## 2. Materials and Experimental Methods

### 2.1. Materials

The experimental material utilized in this study comprised an HDR duplex stainless steel pipe and welding wire produced by Shanghai Shisheng Technology Development Co., Ltd. (Shanghai, China). The chemical compositions of the welding wire and stainless steel pipe are shown in Table 1.

### 2.2. Welding

Two passes of TIG (Tungsten inert gas) welding were adopted, and the welding machine utilized was a WSE-315-1 AC/DC pulsed argon arc welding machine (Shanghai, China). The HDR steel pipe was halved along the vertical direction of the diameter, and a 60° groove was introduced at the welding position, thus ensuring the absence of blunt edges, as shown in Figure 1.

The connection samples consisted of a total of 7 groups, with the sample numbers and corresponding process parameters being outlined in Table 2. Among them, samples 1 to 3 have varying priming welding currents, while samples 2, 4, and 5 have different covering welding currents. Additionally, samples 2, 6, and 7 exhibit differences in welding gap sizes. The interlayer temperature was measured using a thermocouple method.

### 2.3. Electrochemical Specimen Preparation

On the opposite side of the test surface, copper wires were attached and encapsulated with epoxy resin. The test surface was then meticulously sanded using water sandpaper, gradually progressing to a fineness of 2000 grit. Subsequently, the surface was washed with deionized water and thoroughly dried via air blowing. Finally, the prepared samples were placed in a dry dish ready for further testing.

### 2.4. Microstructure Analysis

An AURIGA scanning electron microscope (SEM) was used to analyze the microstructure, whereas IPP software (version 2.0) was used to calculate the austenite content. The acceleration voltage of the scanning electron microscope was 21 kV, and the working distance was approximately 10 mm. An energy dispersive spectrometer (EDS) was used to analyze the composition of ferrite and austenite in the sample. The microstructure analysis sampling points, as illustrated in Figure 2, include the weld, HAZ, and base metal, denoted by WM, HAZ, and BM, respectively. Here, we focus on the microstructure and properties of the HAZ.

### 2.5. Polarization Curve Test

Following the guidelines outlined in GB/T 17899-1999 [9], the potentiodynamic polarization curve of the HAZ was measured to assess the pitting resistance of HDR duplex stainless steel. The polarization curve was measured using a CHI604E electrochemical workstation (CH Instruments, Inc., Bee Cave, TX, USA) with a three-electrode system, using a platinum electrode as the auxiliary electrode, and a saturated calomel electrode (SCE) as the reference electrode. The test system consisted of a 3.5% NaCl solution, the scanning range was −0.1 V~1.5 V (vs. OCP (open-circuitpotential)), the scanning rate was 0.33 mV/s, and the potential corresponding to a polarization current density of 100 μA/cm^2^ was taken as the pitting potential E_b_.

### 2.6. Electrochemical Impedance Spectroscopy (EIS)

Electrochemical impedance spectroscopy (EIS) provides insight into the characteristics of the passive film formed on the surface of stainless steel. The test solution used was 3.5% NaCl, the test temperature was 30 °C, the test instrument used was a CHI604E electrochemical workstation, the potential of the perturbation signal was 10 mV, and the frequency was 100 kHz~10 mHz. The samples were ground, polished, cleaned, and dried before use.

### 2.7. Double Loop Potentiodynamic Reactivation Test (DL-EPR)

The intergranular corrosion resistance of the HDR steel welding joint in the HAZ was tested using the double-ring potentiodynamic reactivation method in GB/T 4334-2020 [10]. The test solution was 2.0 mol/LH_2_SO_4_ + HCl; it was potentiostatically polarized for 5 min at a potential lower than the open circuit potential of 0.9 V in order to remove the passive film that accumulated on the surface of the sample. After the potential was stabilized, the sample was scanned forward to 0.7 V (vs. OCP) at a speed of 1.5 mV/s, and then swept back to the open circuit potential at the same rate.

## 3. Results and Discussion

### 3.1. Microstructure Observation

The microstructure and morphology of the HAZ is shown in Figure 3. When observing duplex stainless steel under a scanning electron microscope, it can be observed that the grain distribution of ferrite and austenite is interlaced. Ferrite usually appears bright white, while austenite appears darker in color. Through metallographic analysis, information such as the grain size, phase ratio, and grain boundary characteristics of duplex stainless steel can be evaluated.

Based on the data presented in Figure 3, the austenite content of the HAZ was determined under different welding heat inputs using Image Pro Plus (IPP) software. IPP can be used to analyze photos obtained from metallographic microscopy (OM), scanning electron microscopy (SEM), or transmission electron microscopy (TEM), allowing us to obtain the proportions of each component or phase. The results are shown in Table 3.

As indicated in Table 3, specimens 1#, 2#, and 3# exhibit a gradual increase in the priming weld current, resulting in a corresponding increase in austenite content. This observation suggests that higher priming weld currents can promote the formation of austenite in the HAZ. When maintaining a consistent covering welding current, a gradual increase in the cover welding current of samples 4#, 5#, and 2# is observed, resulting in a corresponding increase in the austenite content. The higher the heating temperature, the faster the formation rate of austenite, the shorter the transformation incubation period, and the shorter the transformation end time, correspondingly. The nucleation rate and growth rate of austenite increase, and its content also increases.

With an increase in welding gaps for samples 6#, 2#, and 7#, there is a corresponding gradual increase in the austenite content within the HAZ. This observation suggests that appropriately increasing the welding gap can facilitate the formation of austenite. As the temperature of austenite formation increases, the pushing speed of the austenite phase boundary towards ferrite is greater than that towards cementite. At the same time, the degree of imbalance in the phase transformation increases. At the moment when the ferrite phase disappears, the remaining amount of cementite increases, resulting in a lower average carbon content in austenite.

### 3.2. Polarization Curve Analysis

The anodic polarization curve of the samples was measured using potentiodynamic scanning, and the scanning was stopped when the current density reached 1 mA/cm^2^. The polarization curve of the HAZ of the HDR steel welded joint is shown in Figure 4. The potential corresponding to a polarization current density of 100 μA/cm^2^ is taken as the pitting potential E_b_ of the sample. The E_b_ values obtained from the polarization curve are shown in Table 4.

The nature of metal has an important impact on the pitting sensitivity. The higher the pitting potential of the material, the stronger the pitting resistance. Table 4 shows that the covering welding current of samples 1#, 2#, and 3# increases sequentially, whereas the average E_b_ increases. Consequently, the pitting resistance is improved. Hence, it can be concluded that increasing the covering welding current increases the austenite content, promotes a more uniform structure, and enhances the material’s resistance to pitting corrosion.

The cover welding current of samples 4#, 5#, and 2# increases sequentially. The E_b_ (pitting potential) decreases first and then increases, whereas the pitting resistance does not improve significantly. From the perspective of structure and composition, the covering current has little effect on the distribution of alloy elements. Therefore, to achieve optimal pitting corrosion performance, a covering welding current of 70 A and a cover welding current of 100 A are reasonable welding parameters in terms of E_b_.

### 3.3. Experimental Analysis of Electrochemical Impedance Spectroscopy

The samples were soaked in 3.5% NaCl solution at 30 °C. A sine wave with an amplitude of 10 mV was selected as the excitation signal and was tested over a frequency range of 100 KHz to 10 MHz, scanning from high to low frequency. The AC impedance of the base metal and weld were measured after immersion for 24 h. Then, the stability of the passive films on the surface of the base metal and weld were compared and analyzed. The results are shown in Figure 5.

The equivalent circuit shown in Figure 6 was used to fit the EIS spectrum. In the equivalent circuit diagram, R_s_ represents the solution resistance, R_ct_ represents the charge transfer resistance, CPE represents double the layer capacitance, and n is the dispersion coefficient. The fitted parameters are shown in Table 5.

The nature of the metal has an important impact on the pitting sensitivity. The covering welding current of samples 1#, 2#, and 3# increases sequentially. Moreover, the charge transfer resistance (Rct) increases, improving the corrosion resistance. Increasing the covering welding current increases the austenite content, promotes a more uniform structure, and increases the material’s corrosion resistance. The cover welding currents of samples 4#, 5#, and 2# increase sequentially, and the charge transfer resistance (R_ct_) decreases first and then increases.

The charge transfer resistance (R_ct_) of samples 4# and 7# is high. The primary form of corrosion in duplex stainless steel is pitting corrosion, and the resistance to pitting depends on the ferrite phase, which typically has a weaker resistance to pitting [11,12,13]. The microstructure analysis also reveals that samples 4# and 7# possess higher austenite contents, closely resembling the ferrite content, thus indicating relatively enhanced corrosion resistance in these samples.

### 3.4. Analysis of the Double-Loop Kinetic Potential Reactivation Test

The intergranular corrosion resistance of HDR steel welding joints in the HAZ was evaluated using the double-ring potentiodynamic reactivation method. The intergranular corrosion resistance of the sample was evaluated via measuring the maximum currents (I_a_ and I_r_) of the anodized and reactivated rings. The intergranular corrosion resistance index (R_a_) was calculated using the formula R_a_ = (I_r_/I_a_) × 100%. The test curve is shown in Figure 7.

According to the DL-EPR curve of the HAZ, the R_a_ values under different welding currents were determined, and the results are shown in Table 6.

Table 6 shows that the covering welding current of samples 1#, 2#, and 3# increases sequentially. The R_a_ value gradually increases, weakening the intergranular corrosion resistance. This indicates that increasing the covering welding current is not conducive to the intergranular corrosion resistance of the material. Consequently, the sensitivity to intergranular corrosion in the HAZ area increases sequentially for samples 4#, 5#, and 2#. This results in a fluctuation in the Ra value, initially decreasing before then increasing again. Overall, the intergranular corrosion resistance exhibits no significant improvement. From the perspective of structure and composition, during the transformation from ferrite to austenite, the covering current facilitates the precipitation at the grain boundary, thus reducing its intergranular corrosion resistance [14,15,16]. Hence, an appropriate reduction of the welding current is expected to improve the stability of the grain boundary. Consequently, for the optimal intergranular corrosion resistance, a covering welding current of 65 A and a cover welding current of 95 A are assumed to be reasonable welding parameters.

## 4. Conclusions

(1) The priming welding current can promote the formation of austenite in the HAZ. The correspondence between the cover welding current and austenite formation does not strictly align with the appropriate increase in the welding gap to promote the formation of austenite.

(2) In terms of enhancing pitting resistance, increasing the covering welding current leads to a stronger resistance against pitting. Therefore, setting the covering welding current to 70 A and the covering welding current to 100 A are considered to be reasonable welding parameters.

(3) In terms of the intergranular corrosion resistance, increasing the covering welding current is not conducive to the intergranular corrosion resistance of the material. A covering welding current of 65 A, a covering welding current of 95 A, and a welding gap of 2 mm are considered to be reasonable welding parameters.

## Figures and Tables

**Figure 1 materials-17-01986-f001:**
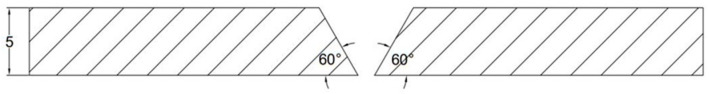
Schematic diagram of the welding groove.

**Figure 2 materials-17-01986-f002:**
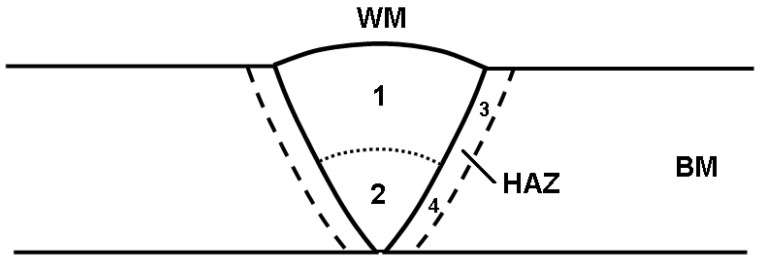
Schematic diagram of the sample for microstructure analysis.

**Figure 3 materials-17-01986-f003:**
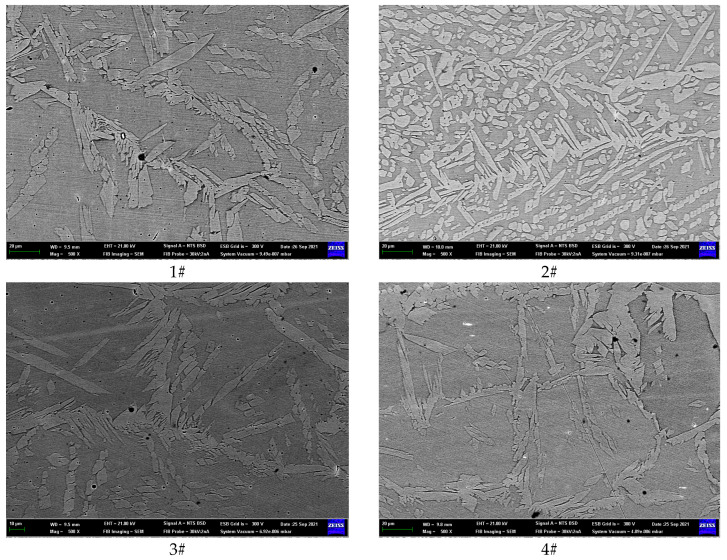
Microstructure of the HAZ of HDR duplex stainless steel.

**Figure 4 materials-17-01986-f004:**
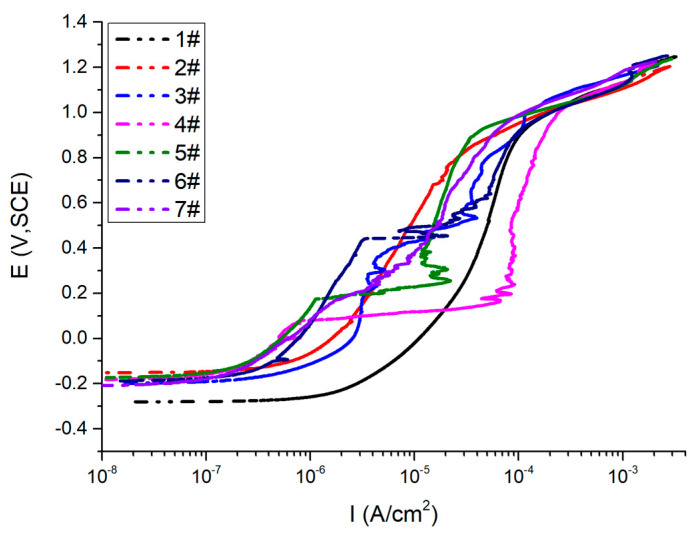
Polarization curves of the HAZ.

**Figure 5 materials-17-01986-f005:**
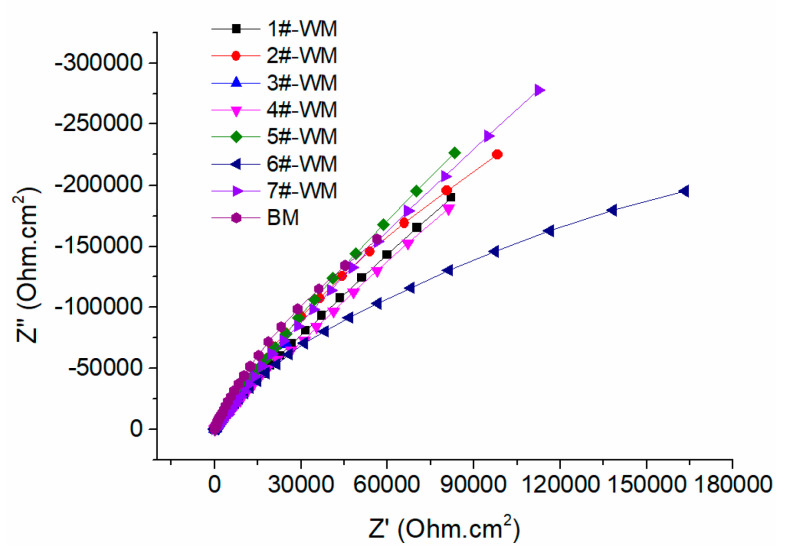
EIS curves of the HAZ (soaked for 24 h).

**Figure 6 materials-17-01986-f006:**
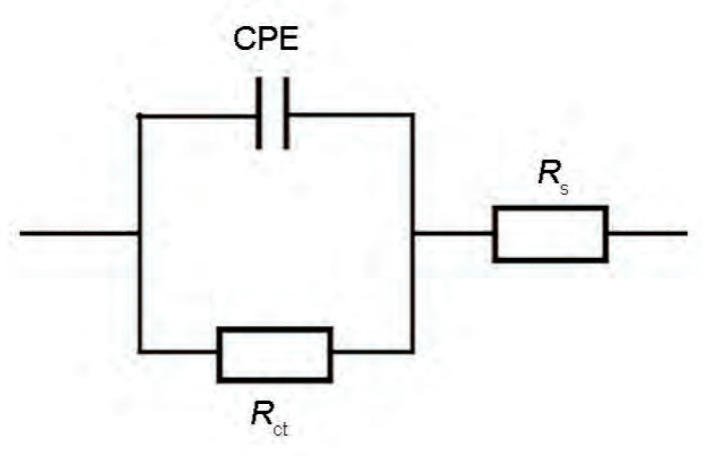
Equivalent circuit diagram of EIS.

**Figure 7 materials-17-01986-f007:**
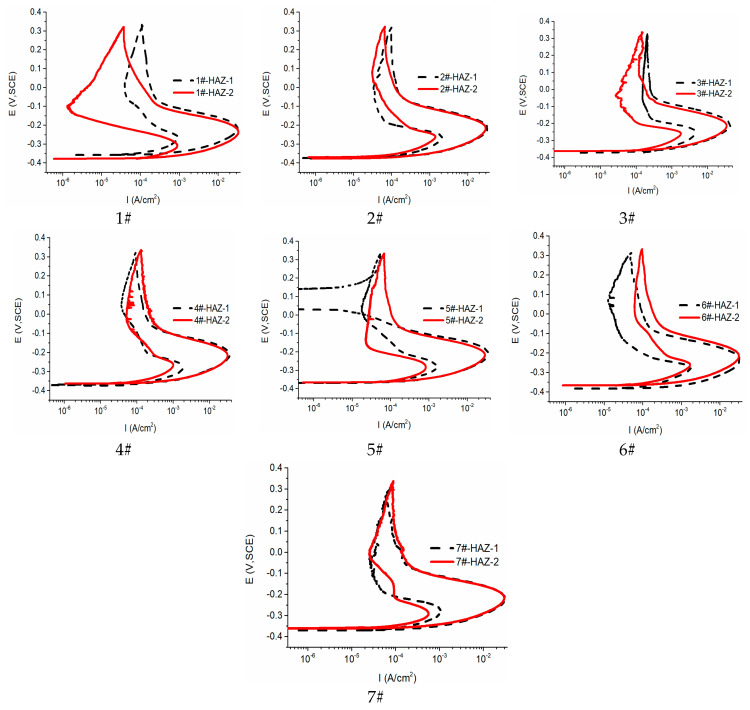
DL-EPR curve of the HAZ.

**Table 1 materials-17-01986-t001:** Chemical compositions of the materials (mass fraction) (wt%).

Elements	Cr	Ni	C	Mo	Si	P	S	N	Mn
HDR	24.54	5.82	0.026	2.15	0.42	0.021	0.001	0.168	1.26
Welding wire	24.73	6.83	0.03	2.09	0.31	0.018	0.005	0.2	1.13

**Table 2 materials-17-01986-t002:** Welding process parameters.

Specimen Number	Weld Passes	Welding Current/A	Interlayer Temperature/°C	Welding Clearance/mm
1#	Priming	65	130–150	2.0
	Covering	100		
2#	Priming	70	130–150	2.0
	Covering	100		
3#	Priming	75	130–150	2.0
	Covering	100		
4#	Priming	70	130–150	2.0
	Covering	90		
5#	Priming	70	130–150	2.0
	Covering	95		
6#	Priming	70	130–150	1.5
	Covering	100		
7#	Priming	70	130–150	2.5
	Covering	100		

**Table 3 materials-17-01986-t003:** Austenite content of the HAZ at different welding heat inputs (%).

No	1#	2#	3#	4#	5#	6#	7#
Content	30.78	36.64	38.40	33.13	34.28	38.06	39.50
Priming current/A	65	70	75	70	70	70	70
Covering current/A	100	100	100	90	95	100	100
Welding clearance/mm	2.0	2.0	2.0	2.0	2.0	1.5	2.5

**Table 4 materials-17-01986-t004:** Pitting potential of the HAZ.

No.	1#	2#	3#	4#	5#	6#	7#
E_b_ (V, SCE)	0.9124	0.9662	0.9673	0.9445	0.9444	0.9763	0.9312
Priming current/A	65	70	75	70	70	70	70
Covering current/A	100	100	100	90	95	100	100
Welding clearance/mm	2.0	2.0	2.0	2.0	2.0	1.5	2.5

**Table 5 materials-17-01986-t005:** Parameters of fitting of the EIS curve of HAZ.

No.	R_s_ (Ω·cm^2^)	n	CPF,Y_0_ (Ω^−1^·cm^−2^·s^−1^)	R_ct_ (10^4^ Ω·cm^2^)
1#	1.135	0.798	1.24 × 10^−3^	75.57
2#	1.644	0.750	1.55 × 10^−3^	119.2
3#	2.151	0.776	1.36× 10^−3^	142.75
4#	2.615	0.773	1.35 × 10^−3^	567.3
5#	0.7868	0.779	1.33 × 10^−3^	78.9
6#	1.597	0.797	1.24 × 10^−3^	41.97
7#	1.679	0.792	1.15 × 10^−3^	197.5

**Table 6 materials-17-01986-t006:** R_a_ values of the HAZ.

No.	1#	2#	3#	4#	5#	6#	7#
R_a_	2.77%	4.05%	5.20%	4.22%	3.70%	5.37%	3.36%

## Data Availability

Data are contained within the article.

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
