# Peer review of "Effect of Welding Current on Corrosion Resistance of Heat-Affected Zones of HDR Duplex Stainless Steel"

_materials, 2024, doi:10.3390/ma17091986_

Round 1
Reviewer 1 Report
Comments and Suggestions for Authors
Authors need to revise the manuscript, as per the comments added in the annotated pdf file attached herewith. Thnak you.

Reviewer 2 Report
Comments and Suggestions for Authors
The authors write "the polarization current density was 100μ". What does it mean ??? The figures show that the measurement was completed at 100 μA. Please describe it clearly in the research methodology.
The authors provide a criterion for determining the pitting corrosion potential as "The potential corresponding to A/cm2 corresponds to the pitting potential Eb". What does "A/cm2" mean? Please provide the exact value of the current at which the zero corrosion potential was determined.
"the potential polarization was 10 mV" - this is how DC polarization is defined. In the case of EIS research, we rather use the term "perturbation signal", indicating its alternating current nature.
The authors obtained very high values of zero corrosion potentials. Was pitting corrosion really found in all tested locations? The authors do not show a view of the surface tested for pitting, nor did they perform return curves (cyclic polarization). The shape and orientation of the recovery curve would clearly indicate susceptibility to pitting corrosion (hysteresis).
If a constant-phase CPE element is used in the equivalent circuit, F/cm2 should not be used as the unit. This needs to be improved. (table 5).
Shouldn't the equivalent diagram be simplified to the R(RQ) system?
Round 2
Reviewer 1 Report
Comments and Suggestions for Authors
Authors have satisfactorily addressed most of the comments raised by the reviewer. This manuscript is now accpeted for publication.
Author Response
Thank you for careful review, which has greatly benefited me academically. Thank you!
Reviewer 2 Report
Comments and Suggestions for Authors
Please look carefully for the unit applicable to CPE, because the one presented by the authors is not true.
It is necessary to present hysteresis (or a view of the surface after testing) because with such values of the zero corrosion potential it is not certain whether the tested steel has undergone this type of corrosion.
